# Measurement of the spin-forbidden dark excitons in MoS₂ and MoSe₂ monolayers

C. Robert [1,7 ✉], B. Han [1,7 ✉], P. Kapuscinski [2,3,7], A. Delhomme[2], C. Faugeras [2 ✉], T. Amand[1], M. R. Molas [4], M. Bartos[2,5], K. Watanabe [6], T. Taniguchi[6], B. Urbaszek [1], M. Potemski [2,4] & X. Marie [1]

Excitons with binding energies of a few hundreds of meV control the optical properties of transition metal dichalcogenide monolayers. Knowledge of the fine structure of these excitons is therefore essential to understand the optoelectronic properties of these 2D materials. Here we measure the exciton fine structure of MoS₂ and MoSe₂ monolayers encapsulated in boron nitride by magneto-photoluminescence spectroscopy in magnetic fields up to 30 T. The experiments performed in transverse magnetic field reveal a brightening of the spin-forbidden dark excitons in MoS₂ monolayer: we find that the dark excitons appear at 14 meV below the bright ones. Measurements performed in tilted magnetic field provide a conceivable description of the neutral exciton fine structure. The experimental results are in agreement with a model taking into account the effect of the exchange interaction on both the bright and dark exciton states as well as the interaction with the magnetic field.

[1] University of Toulouse, INSA-CNRS-UPS, LPCNO, 135 Av. Rangueil, 31077 Toulouse, France. [2] Laboratoire National des Champs Magnétiques Intenses, CNRS-UGA-UPS-INSA-EMFL, 38042 Grenoble, France. [3] Department of Experimental Physics, Faculty of Fundamental Problems of Technology, Wrocław University of Science and Technology, Wybrzeże Wyspiańskiego 27, 50-370 Wrocław, Poland. [4] Institute of Experimental Physics, Faculty of Physics, University of Warsaw, ul. Pasteura 5, 02-093 Warsaw, Poland. [5] Central European Institute of Technology, Brno University of Technology, Purkynova 656/123, 61200 Brno, Czech Republic. [6] National Institute for Materials Science, Tsukuba, Ibaraki 305-0044, Japan. [7] These authors contributed equally: C. Robert, B. Han, P. Kapuscinski. ✉email: cerobert@insa-toulouse.fr; bhan@insa-toulouse.fr; clement.faugeras@lncmi.cnrs.fr

Transition Metal Dichalcogenide (TMD) monolayers such as $MoS_2$, $MoSe_2$, $WS_2$ or $WSe_2$ are 2D semiconductors characterized by very strong interaction with light due to robust excitons with large oscillator strengths[1–5]. As for all semiconductor nanostructures the exciton fine structure dictates the efficiency of the coupling to light. One can expect that the optoelectronic properties will change drastically whether the spin-forbidden dark excitons lie below or above the bright excitons[6–9]. Exciton spin relaxation is also expected to be affected by the bright-dark exciton ordering[10,11].

The main difficulty to experimentally determine the energy of spin-forbidden dark excitons is their extremely small oscillator strength compared to the bright exciton one. The exciton fine structure splitting was accurately determined for $WS_2$ and $WSe_2$ monolayers (ML) using various experimental techniques[12–15]. These measurements were successful only in $WSe_2$ and $WS_2$ materials as the dark exciton lie several tens of meV below the bright one so that the small oscillator strength is compensated by a very large population of dark excitons making them observable in photoluminescence experiments. In contrast the respective alignment of bright and dark excitons in $MoS_2$ ML remains controversial though this material is the most studied among the 2D semiconductors and this was the first member of the TMD family to be established as a direct gap in the monolayer form.

Numerous ab-initio calculations have been proposed to predict the exciton bright-dark splitting but the results are highly dispersed with values in the range 10–40 meV and more importantly with different signs: depending on the methods applied, dark excitons lie above or below the dark ones[16–20]. It is therefore crucial to have a clear experimental determination.

Unfortunately, all attempts to measure the bright-dark splitting in $MoS_2$ ML were not conclusive so far. The main reasons given were related to (i) the low optical quality of samples (without hBN encapsulation)[13], (ii) the small value of the expected splitting compared to the luminescence/absorption linewidth or (iii) the low thermal population of the dark states if they lie above the bright ones[12].

In this letter, we present the unambiguous optical emission spectrum of the dark exciton in $MoS_2$ ML. In contrast to most predictions our measurements demonstrate that the spin-forbidden dark exciton lies below the bright exciton with a bright-dark splitting of +14 meV.

To this end, we performed magneto-photoluminescence experiments up to 30 T on $MoS_2$ monolayers encapsulated in hexagonal Boron Nitride (hBN) with a magnetic field oriented along the monolayer plane (Voigt geometry). The high quality of the investigated samples allowed us to determine accurately the bright-dark energy splitting. We performed similar measurements in $MoSe_2$ MLs encapsulated in hBN where a bright-dark splitting of −1.3 meV is determined in agreement with a recent report[21]. Measurements performed in a tilted magnetic field (45° with respect to the ML plane) confirm the spin-forbidden nature of the observed transitions and allow us to provide a conceivable description of the neutral exciton fine structure. The in-plane field component yields a brightening of the dark states and the out-of plane component induces a Zeeman spitting of these states yielding the first measurements of the dark exciton g-factor in $MoS_2$ and $MoSe_2$ monolayers. Remarkably, we find that the energy splitting between the bright and dark excitons in $MoSe_2$ ML is very similar to the short range exchange energy which splits the two dark exciton states, a very original and unique situation for semiconductor nanostructures.

## Results

**Mixing of exciton states in magnetic field.** First, let us recall that the point symmetry group of a TMD monolayer is $D_{3h}$. The direct band gap is located at the edges of hexagonal Brillouin zone, at the non-equivalent valleys $K_\pm$. Because of the large spin–orbit splitting in the valence band, we restrict here our description to A excitons composed of an electron from one of the two conduction bands split by the spin–orbit interaction $\Delta_{SO}$ and a hole from the upper valence band A. We also consider only direct excitons (with a center of mass wave-vector $K = 0$). Then the exciton fine structure includes the two optically active (bright) excitons $X_B$ with parallel spin ↑↑, ↓↓ (symmetry $\Gamma_6$) and the two spin-forbidden dark states $X_G(↑↓ + ↑↓)$ and $X_D(↑↓−↑↓)$ (symmetry $\Gamma_4$ and $\Gamma_3$ respectively). The inset of Fig. 1c shows the bright-dark splitting $\Delta$ and the gray-dark splitting $\delta$ due to exciton exchange energy. It was shown that these dark states are optically forbidden for in-plane polarized light but that the $X_G$ ($\Gamma_4$) state can couple to out-of plane polarized light (Oz direction)[12,22,23] and because of this is called here "gray" exciton.

An in-plane magnetic field $\mathbf{B}_{//}$ (Voigt geometry) mixes the spin components of the 2D exciton states[24,25]. For a TMD monolayer, this field interacts with the conduction band electron within the exciton and we can neglect its interaction with the hole because of the very large valence band spin–orbit splitting[13,14]. As a consequence the in-plane magnetic field brightens both $X_G$ and $X_D$ dark exciton states; the mixed bright-dark exciton states couple to in-plane polarized light allowing a straightforward determination of the energy of the dark states[13,14]. On the other hand, an out-of plane magnetic field $\mathbf{B}_z$ (Faraday geometry) leads to the Zeeman splitting of both bright and dark states and to the mixing between $X_G$ and $X_D$[22]. Thus, in tilted magnetic field, the four excitons are mixed and split so that it becomes possible to extract the g-factor of dark excitons. The $(4 \times 4)$ Hamiltonian describing the mixing between the four states under a given field with an angle $\theta$ with respect to the normal of the ML plane ($\theta = 90°$ for Voigt and $\theta = 0°$ for Faraday geometries) is described in the Supplementary Note 6.

**Magneto-photoluminescence for in-plane magnetic field.** First we have investigated the effect of an in-plane magnetic field (Voigt configuration) on the low-temperature photoluminescence spectra in $MoS_2$ monolayer. The 2D color map of PL intensity as a function of magnetic field from 0 to 30 T is plotted in Fig. 1a. In zero field, the emission is composed of a unique line corresponding to the radiative recombination of the bright exciton $X_B$ at 1.931 eV in agreement with previous reports[26]. Remarkably we observe at low energy, typically 14 meV below $X_B$, an additional peak which shows up above ~12 T. This feature, interpreted as the brightened spin-forbidden dark exciton, has been reproduced on several samples and spot positions (see data in Supplementary Note 1 principle), this line should correspond to both brightened gray and dark excitons, but the inhomogeneous linewidth in our samples is too broad (the linewidth of $X_D$ is 5 meV) to enable us to distinguish the expected small splitting $\delta$ between gray and dark states. In a simple two-level system ($\delta = 0$) where the in-plane magnetic field couples the bright and dark states, one expects that the ratio between the PL intensity of the bright and the dark PL lines follows a simple quadratic law: $I_D/I_B \sim (\mathbf{B}_{//})^2$, where $\mathbf{B}_{//}$ is the amplitude of the in-plane magnetic field[14]. In Fig. 1c, we present the magnetic field dependence of the ratio between the PL intensity of the low-energy and the high-energy lines (corresponding to exciton states dominated by dark and bright components, respectively). The measured quadratic behavior is a strong indication that the low-energy line corresponds to

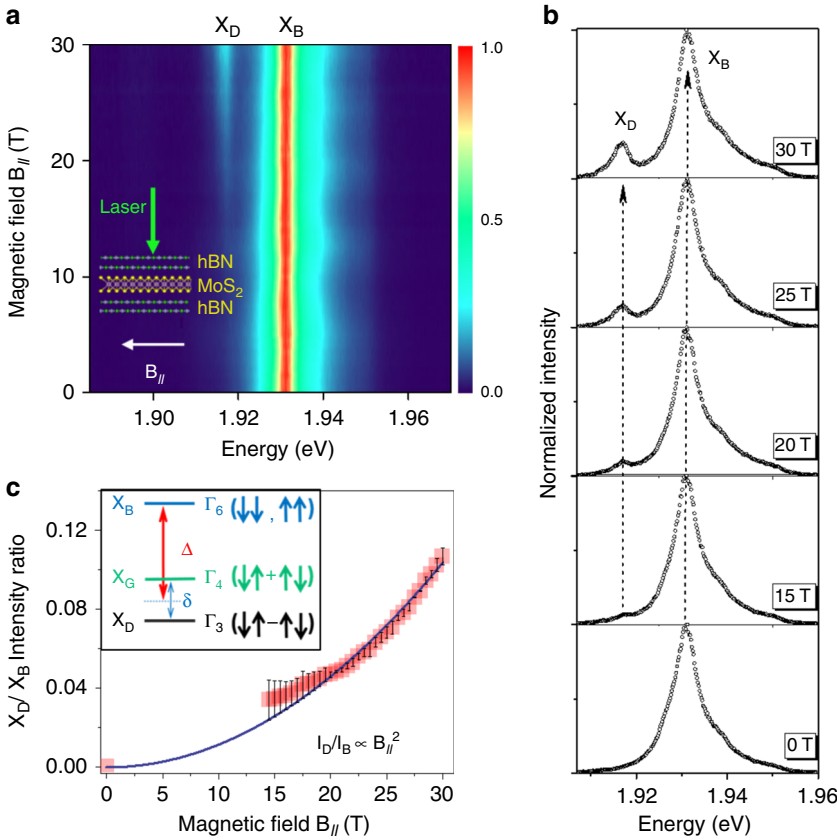

**Fig. 1 In-plane magnetic field $B_{//}$—spin-forbidden dark exciton in MoS$_2$ monolayer encapsulated in hBN revealed by photoluminescence. a** Color map of the variation of the PL intensity as a function of $\mathbf{B}_{//}$ (the PL intensity of the bright exciton has been normalized at each field); **b** PL spectra for magnetic fields from 0 to 30 T showing the emergence of the brightened dark exciton at low energy. **c** Ratio of the PL intensity of dark ($X_D$) and bright ($X_B$) excitons as a function of magnetic field. Inset: sketch of the excitonic fine structure. The arrows ↑ and ↓ represent the main spin contribution of conduction and valence electrons involved in the exciton states (see ref. [22] for more details).

| | MoS$_2$ | MoSe$_2$ | WSe$_2$ | WS$_2$ |
|---|---|---|---|---|
| **Table 1 Measured exciton fine structure parameters for hBN-encapsulated TMD monolayers.** | | | | |
| Splitting between bright and dark exciton $\Delta$ (meV) | +14 | −1.4<br>−1.5[21] | +40[12,15] | +55[12]<br>+40[42] |
| Splitting between gray and dark excitons $\delta$ (meV) | <2 | <1 | 0.6[22,28] | 0.5[42] |
| Bright exciton $g$-factor | −1.8 | −4.0 | −4.25[43] | −4.0[29] |
| Dark exciton $g$-factor | −6.5 | −8.6 | −9.4[22,28] | — |
| Transverse electron $g$-factor | 2 | 2<br>2[21] | 2[28] | — |

the recombination of the dark excitons which have been brightened by the application of the magnetic field. An additional evidence for assigning the low-energy line observed for large in-plane magnetic field to spin-forbidden dark excitons is the measurements of its $g$-factor in an external field perpendicular to the monolayer. The measurements in the tilted field geometry presented below yield $g_z^D = -6.5$; this large value is in good agreement with the predicted one in a simple model[27] (see Supplementary Note 9) and the measured one in WSe$_2$ MLs[22,28] (Table 1).

We have also evidenced the energy of the spin-forbidden dark exciton in MoSe$_2$ MLs encapsulated in hBN using the same experimental approach. As shown in the 2D color map of the PL spectra in Voigt configuration presented in Fig. 2a, the brightened dark exciton line (labeled for simplicity $X_D$) lies above the bright

exciton one ($X_B$) in contrast to MoS$_2$ (compare with Fig. 1a). Moreover, the brightened dark exciton starts to be visible at much lower field (~8 T) and the energy of the two lines vary much strongly with $\mathbf{B}_{//}$ as a consequence of the much smaller bright-dark splitting $\Delta$. Again, the linewidth of the transitions (1.4 meV for $X_B$ and 2.4 meV for $X_D$) does not allow us to observe the splitting $\delta$ between gray and dark excitons. Using the general Hamiltonian presented in the Supplementary Note 6, including the effect of the exciton exchange interaction and the external magnetic field (with arbitrary orientation $\theta$ with respect to the ML plane), we can easily calculate the $\mathbf{B}_{//}$ field dependence of the bright, gray and dark exciton energies. Taking $\theta = 90°$ (Voigt geometry) and $\delta = 0.6$ meV, we can fit our data (black solid lines in Fig. 2b) with $\Delta = -1.4 \pm 0.1$ meV and $g_{//} = 2.0 \pm 0.2$. These values are in good agreement with very similar measurements published very recently[21]. Note that the fit is very weakly sensitive to the value of $\delta$ for $\delta < 1$ meV. Supplementary Figure 9a shows the results of the fit with three values of $\delta$ (0, 0.6, and 2 meV). In Fig. 2b, we present the results of the fit for $\delta = 0.6$ meV which corresponds to the value experimentally measured in WSe$_2$[22,28]. As $\delta$ is due to short range exchange interaction, which scales with the exciton binding energy, we do not expect the values to be strongly different between TMD materials as the exciton binding energies are roughly the same[29].

In MoTe$_2$ ML encapsulated with hBN, our measurements did not reveal any new PL line at high magnetic field that could have been attributed to spin-forbidden dark excitons. Our interpretation is that these excitons have a higher energy compared to the bright ones. Thus, a very small thermal population of dark states

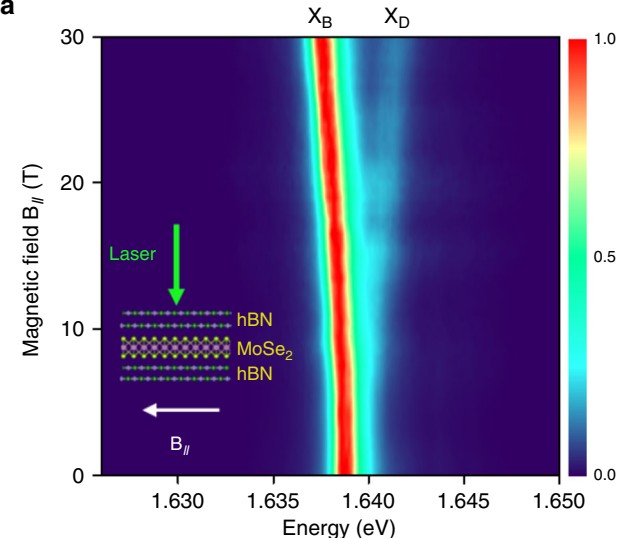

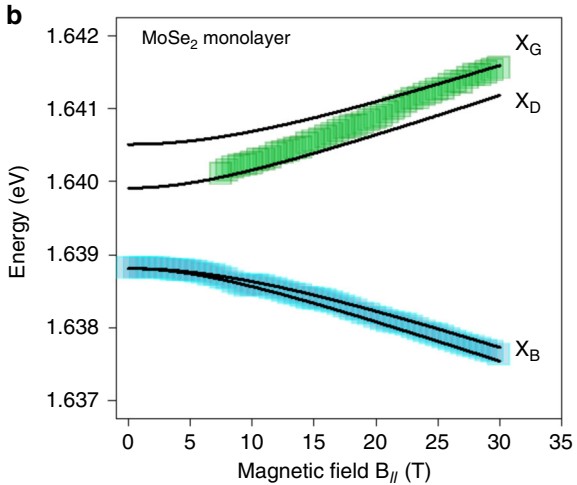

**Fig. 2 In-plane magnetic field $B_{//}$—spin-forbidden dark exciton in MoSe$_2$ monolayer encapsulated in hBN revealed by photoluminescence. a** Color map of the variation of the PL intensity as a function of $\mathbf{B}_{//}$ (the PL intensity of the bright exciton has been normalized at each field); **b** energy of dark and bright excitons as a function of $\mathbf{B}_{//}$. The solid lines correspond to the fit described in the text.

is expected and a much larger magnetic field is required to mix them with bright excitons. This is consistent with the fact that the calculated CB spin–orbit splitting is twice as large as for MoTe$_2$ as for MoSe$_2$[19].

**Magneto-photoluminescence in tilted magnetic field.** In order to have a full description of the exciton fine structure, we have measured the PL spectra of MoS$_2$ and MoSe$_2$ monolayers in a tilted magnetic field configuration (the field is oriented 45° with respect to the 2D layer plane). These experiments allow the determination of several key parameters, such as the dark exciton g-factors. In an oversimplified description, one can consider separately the effect of the two field components on the exciton spectra. The in-plane component of the field yields the mixing of the bright and dark exciton as already observed in the Voigt configuration and the out-of plane component leads to a Zeeman splitting of the states. While the energy splitting between the two spin/valley states for the bright exciton depends linearly with this z-component field (the slope given by effective g-factor), the field

dependence of the dark states is more complicated since the two dark states at zero field are split by the exchange energy $\delta$ of the order of a few hundreds of $\mu eV$[22,28] (inset of Fig. 1c).

First we present the dependence of the PL spectra of MoSe$_2$ monolayer in tilted magnetic field. At high magnetic field (above ~12 T), the color map of the PL intensity in Fig. 3a and the PL spectra in Fig. 3b clearly evidence 4 lines corresponding to the 4 exciton states whose energy vary almost linearly with the field in the range 15–30 T (i.e., the out-of plane component varying from ~10 to 21 T). For lower magnetic field values, the energy splitting between the states is comparable to the PL linewidth preventing an accurate determination of the energy of the 4 states. However, we observe a clear non-linear dependence of the energy of the main PL line (which corresponds at $B = 0$ to the bright exciton) (Fig. 3c). This is a consequence of both the effect of the zero-field bright-dark splitting $\Delta$ and the zero-field splitting between the gray and dark states $\delta$.

Thus, the rigorous description requires to consider both the exciton interaction with the tilted magnetic field $\mathbf{B} = \mathbf{B}_{//} + \mathbf{B}_z$ and the exciton exchange interaction; $\mathbf{B}_{//} = B\sin\theta$ and $\mathbf{B}_z = B\cos\theta$ are the magnetic field components parallel and perpendicular to the monolayer plane (in the experiments of Fig. 3, $\theta = 45°$). The interaction with the magnetic field is driven by $g_z^B$ and $g_z^D$, which are respectively the exciton g-factor of bright and dark excitons and $g_{//}$, which is the in-plane electron g-factor. The eigenstates for each magnetic field value have been obtained numerically (Supplementary Note 6). On the basis of this model and using the values $\Delta = -1.4\,meV$ and $g_{//} = 2$ obtained from the Voigt experiments (Fig. 2), one can fit simultaneously the field dependence of the energy of the 4 lines presented in Fig. 3c (see the solid line for the calculated curves). The agreement between the experiment and theory is very good. Interestingly, a clear anti-crossing is evidenced in the low field region as a consequence of the interplay of the transverse and longitudinal field components. This fitting procedure yields the g-factor of both the bright and dark states: we find $g_z^B = -4.0$ and $g_z^D = -8.63$. The value of the dark exciton g-factor around $-9$ is very similar to WSe$_2$ ML[22,28]; it is an additional proof of the highlighting of the spin-forbidden dark states. As we cannot extract the energy of the 4 lines at weak magnetic field due to their linewidths, the fit is not very sensitive to the zero-field splitting $\delta$ between the gray and dark states (the curves have been calculated for $\delta = 0.6\,meV$, the value measured in WSe$_2$ ML[22,28]). Supplementary Figure 9b shows the results of the fit with three values of $\delta$ (0, 0.6, and 2 meV). Note that the labeling of the 4 lines ($X_B$, $X_B$, $X_D$, $X_G$) in Fig. 3 is only valid at zero field. When $\mathbf{B} \neq 0$, the 4 states are mixed. In Fig. 3d, we show the mixing of each state as a function of $\mathbf{B}$. In Supplementary Note 7, we show the calculation of the weight of the 4 components at 30 T.

Finally, we have measured the excitons spectra of MoS$_2$ ML in tilted magnetic field. Due to the larger PL linewidth compared to the one of MoSe$_2$ ML, it is more difficult to evidence the 4 excitons states as in Fig. 3a. However the energy of the two Zeeman split dark states can be extracted above 15 T as shown in Fig. 4a–c displays the measured magnetic field dependence of the energies of the four states together with the fit based on the same model as MoSe$_2$. Using $\Delta = +14.0\,meV$ and $g_{//} = 2$ (determined previously in the Voigt geometry), the best fit is obtained for $g_z^B = -1.8$ and $g_z^D = -6.5$. The bright exciton g-factor of about $-2$ was already measured in high-quality MoS$_2$ monolayer[30]. However, this is here the first measurement of the dark exciton g-factor. Interestingly, we note that the g-factors of both bright and dark excitons are significantly smaller in MoS$_2$ than in other TMD materials. We can speculate that this is due to the very small conduction band spin–orbit splitting. Similarly, to MoSe$_2$,

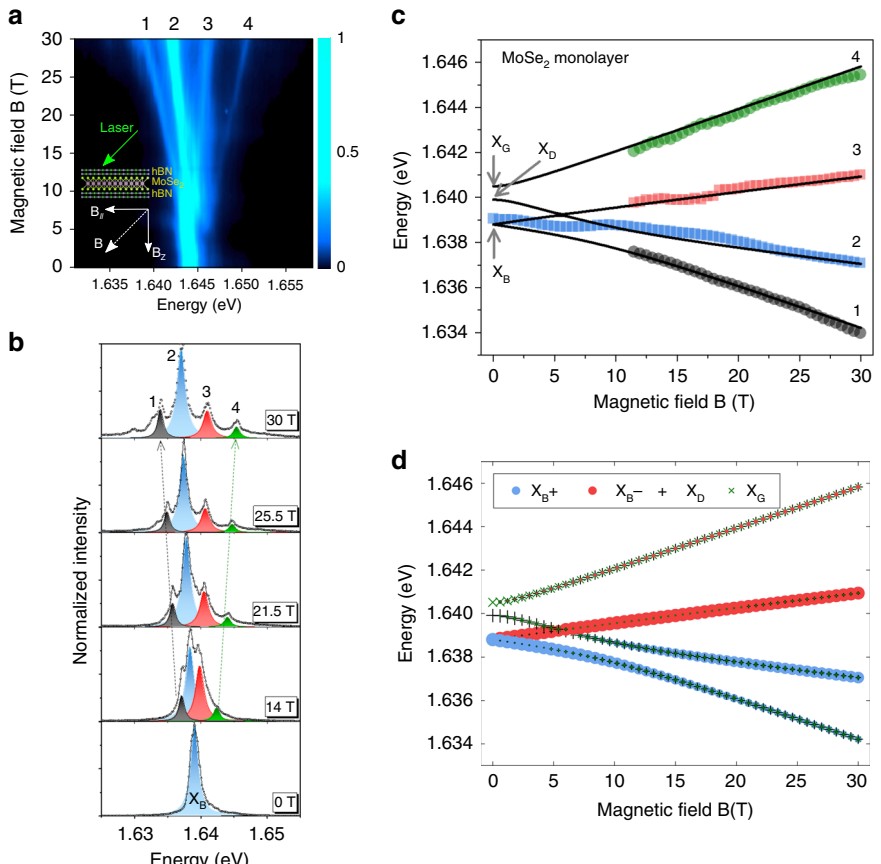

**Fig. 3 Tilted (45°) magnetic field—the four mixed excitons states (labeled 1, 2, 3, 4) in MoSe₂ monolayer revealed by magneto-photoluminescence.** **a** Color map of the variation of the PL intensity as a function of **B**; **b** PL spectra for magnetic fields from 0 to 30 T showing the emergence of the four mixed states (the intensity is normalized to the main exciton line (state 2)). **c** Magnetic field dependence of the energy of the four mixed exciton states. The full lines are fits to the model described in the text. The notations dark exciton ($X_D$), gray exciton ($X_G$) and bright excitons ($X_B$) are only strictly valid at 0 T. **d** Same fitting results than in **c**. The size of each symbol is proportional to the weight of each component (the two bright component $X_B+$ and $X_B-$ and the two dark component $X_G$ and $X_D$) (see Supplementary Note 7 for the calculation of the weight of the four components at high field).

the fit is not sensitive to the value of the dark-gray splitting due to the lack of experimental data at low field (the calculation has been done for $\delta = 0.6$ meV). Supplementary Figure 9c shows the results of the fit with three values of $\delta$ (0, 0.6, and 2 meV). Table 1 summarizes the parameters of the exciton fine structure of MoS₂ and MoSe₂ monolayers presented in this Letter, together with the ones of WS₂ and WSe₂ MLs from previous work.

## Discussion

The measured value of $\Delta = 14$ meV in MoS₂ ML generates several remarks and interrogations.

(i) First it demonstrates the key role played by the exciton exchange energy contribution to the bright-dark energy splitting. The bright-dark exciton splitting $\Delta$ includes three contributions[17–19]:

$\Delta = \Delta_{SO} + \Delta_{bind} + \Delta_{exch.}$, where $\Delta_{SO}$ is the conduction band spin–orbit splitting, $\Delta_{bind}$ is the difference between the binding energies of bright and dark excitons due to the slightly different masses of spin ↑ and spin ↓ conduction bands and $\Delta_{exch}$ is the short range exciton exchange energy. Although the actual value of the spin–orbit splitting is not yet known, the most widely used calculated value in the literature is $\Delta_{SO} = -3$ meV[31]. In a first approximation, the binding energy is proportional to the reduced mass of the exciton. Taking the calculated effective masses from ref. [31] and the experimental value of the bright exciton binding energy from ref. [29], we infer $\Delta_{bind} = +8$ meV. As a consequence,

our measurement of $\Delta = +14$ meV demonstrates that the exciton exchange energy is crucial to determine the amplitude and the sign of the bright-dark energy splitting. Thus we deduce $\Delta_{exch} \sim +9$ meV (if indeed $\Delta_{SO} = -3$ meV and $\Delta_{bind} = +8$ meV). Note that the CB spin–orbit splitting in MoS₂ monolayer has been recently estimated from transport measurements[32]; a splitting of about 15 meV was measured for an electron density of a few $10^{12}$ cm⁻². This value five times larger than the calculated one includes significant band renormalization induced by many body effects.

(ii) Importantly, our measurements show that the splitting between spin-forbidden bright and dark excitons in MoS₂ ML has an opposite sign compared to the calculated spin–orbit splitting in the conduction band. This could have important consequences for the trion fine structure[33].

(iii) Our measurements raise the question of the simple interpretation of MoS₂ as a bright material. Indeed the TMD monolayers are usually divided into two categories: the so-called dark materials such as WS₂ and WSe₂ MLs, where the spin-forbidden dark excitons lie at lower energy compared to the bright ones. As a consequence, these monolayers are characterized by a rather weak luminescence yield at low temperature while the intensity increases with temperature due to thermal activation of bright states[6–9]. In contrast, MoX₂ monolayer (X = S, Se, or Te) are often considered as "bright" materials as they exhibit stronger PL intensity at low temperature than their W-based counterparts but their intensity drops with temperature.

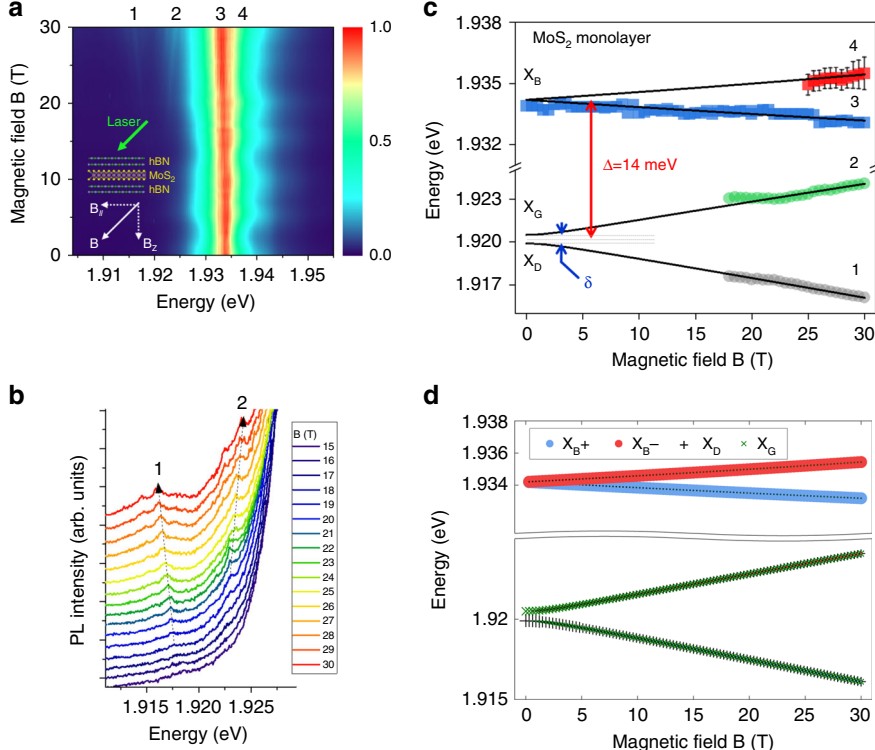

**Fig. 4 Tilted (45°) magnetic field—the four mixed excitons states (labeled 1, 2, 3, 4) in MoS₂ monolayer revealed by magneto-photoluminescence. a** Color map of the variation of the PL intensity as a function of B; **b** PL spectra for magnetic fields from 15 to 30 T showing the emergence of the two lowest energy states 1 and 2 (mainly with gray and dark components). **c** Magnetic field dependence of the energy of the four mixed exciton states. The full lines are fits to the model described in the text. The notations dark exciton ($X_D$), gray exciton ($X_G$) and bright excitons ($X_B$) are only strictly valid at 0 T. **d** Same fitting results than in **c**. The size of each symbol is proportional to the weight of each component (the two bright component $X_B+$ and $X_B-$ and the two dark component $X_G$ and $X_D$) (see Supplementary Note 7 for the calculation of the weight of the four components at high field).

This difference was assumed to vouch for dark excitons lying above the bright ones in MoX₂. This bright-dark ordering has indeed been observed very recently for MoSe₂ ML[21] and confirmed by our results presented above. The measurements displayed in Fig. 1 show that the ordering is surprisingly opposite in MoS₂ ML although the temperature dependence of the PL intensity in MoS₂ ML is very similar to MoSe₂ (strong at low temperature and decrease when temperature increases). Thus, we cannot use this simple argument to distinguish between a bright and a dark material as our results prove that the dependence of PL intensity with temperature in MoS₂ may be the result of a complex relaxation scheme between bright and dark states as well as its interplay with non-radiative channels.

In addition, one can wonder why the gray exciton cannot be detected at zero magnetic field using high numerical aperture objective like in WS₂ and WSe₂[12] (Supplementary Note 4). Remarkably, we note in Fig. 1 that the PL intensity of the mixed dark-bright state in transverse magnetic field is very weak (compared to similar experiments performed in WS₂ or WSe₂ MLs[12,13]) and that very high field are required to sizably observe it (larger than 14 T). We can speculate that either the oscillator strength of gray exciton is much smaller than in WX₂ and/or that their population remains weak despite lying at lower energy than the bright state. We can tentatively explain the small oscillator strength by noticing that the smaller spin–orbit interaction in MoS₂ compared to WS₂ or WSe₂ may yield a weaker oscillator strength of the gray exciton since it is related to the spin–orbit mixing with higher energy bands[12,19]. For the population issue, we can notice that contrary to WX₂, the bright-dark exciton splitting in MoS₂ ML is smaller than the optical phonon energies

which can lead to inefficient relaxation between bright and dark excitons (the $\Gamma_5$ phonon that permit relaxation from bright to dark exciton is 36 meV)[3,34].

Another striking difference between TMD materials is the amplitude of spin/valley polarization for excitons. Although strong valley polarization and valley coherence have been measured in WSe₂, WS₂, and MoS₂ MLs, the polarization in MoSe₂ and MoTe₂ MLs is very weak[35,36] except under quasi-resonant excitation where significant polarization has been measured for the trion in MoSe₂[37]. Recently, it was proposed that a crossing of bright and dark exciton dispersion curves combined with a Rashba effect associated with local fluctuations of electric field can lead to very fast spin relaxation[11] . Our measurements of bright-dark splitting in MoS₂ and MoSe₂ are perfectly consistent with this scenario: the small negative splitting $\Delta = -1.4$ meV in MoSe₂ combined with a larger effective mass for dark excitons should lead to a crossing between bright and dark dispersions while the positive splitting $\Delta = +14$ meV in MoS₂ guarantees no crossing and as a consequence significant spin/valley exciton polarization measured in MoS₂ MLs under CW optical orientation experiments[38–40].

In conclusion, we have performed magneto-photoluminescence experiments in transverse and tilted magnetic fields up to 30 T in high-quality MoS₂ and MoSe₂ monolayers. These investigations yield the unambiguous determination of the bright-dark exciton splitting and the dark exciton g-factor. Such fundamental parameters are key elements to understand the optoelectronic and spin/valley properties of these 2D semiconductors as well as their associated van der Waals heterostructures.

## Methods

**Sample fabrication**. We fabricated high-quality samples by encapsulating $MoS_2$ and $MoSe_2$ MLs in hexagonal boron nitride (hBN). The heterostructures are fabricated onto $SiO_2$ (80 nm)/Si substratesusing a dry stamping technique[12,41]. The typical thickness of the top (bottom) hBN layer is ~10 (200) nm and the typical in-plane size of the ML is ~$10 \times 10 \, \mu m^2$.

**Experimental setup**. Low-temperature magneto-PL experiments are performed in the Voigt configuration (magnetic field parallel to the layer plane) or tilted configuration (field oriented 45° with respect to the ML plane) using an optical fiber-based insert placed in a resistive solenoid producing magnetic fields up to 30 T. The samples are placed on top of an $x$–$y$–$z$ piezo-stage kept in gaseous helium at $T =$ 4.2 K. The light from a cw 515 nm laser is coupled to a mono-mode optical fiber with a core diameter of 5 μm and focused on the sample by an aspheric lens (spot diameter around 2 μm). The PL signal is collected by the same lens, injected into a multi-mode optical fiber of 50 μm core diameter, and analyzed by a 0.5 m long monochromator equipped with a charge-coupled device (CCD) camera. A sketch of the setup is shown in the Supplementary Methods.

**Reporting summary**. Further information on research design is available in the Nature Research Reporting Summary linked to this article.

## Data availability

The data that support the findings of this study are available from the corresponding author upon request.

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

## Acknowledgements

We acknowledge the discussions with A. Slobodeniuk and funding from ANR 2D-vdW-Spin, ANR VallEx, ANR MagicValley, ANR Graskop (ANR-19-CE09-0026) projects, Labex NEXT projects VWspin and MILO, EU Graphene Flagship project (No. 785219), ATOMOPTO project (TEAM programme of the Foundation for Polish Science, co-financed by the EU within the ERD Fund), National Science Centre, Poland (Grant No. UMO-2018/31/B/ST3/02111 and 2016/23/G/ST3/04114), and ME-YS-Czech Republic CEITEC 2020 (LQ1601) project. K.W. and T.T. acknowledge support from the Elemental Strategy Initiative conducted by the MEXT, Japan and the CREST (JPMJCR15F3), JST. X.M. also acknowledges the Institut Universitaire de France.

## Author contributions

C.R., B.H., P.K., A.D., C.F., and M.R.M. performed the high magnetic field experiments. K.W. and T.T. grew the hBN crystals. B.H. and M.B. fabricated the encapsulated samples. T.A. developed the theory. C.R., X.M., and M.P. suggested the experiment. C.R. and X.M. wrote the manuscript with inputs from B.H., P.K., C.F., T.A., B.U., and M.P.

## Competing interests

The authors declare no competing interests.

**Additional information**

