## [Peer Review File · Nature Communications]

Reviewers' comments:

Reviewer #1 (Remarks to the Author):

In this work, Robert et al. studied the PL from monolayer MoS₂ and MoSe₂ in the in-plane magnetic field or tilted magnetic field. They found an emerging peak in MoS₂ and MoSe₂ with the in-plane magnetic field, which they attribute to the dark exciton. Further application of the tilted magnetic field also enabled authors to extract the g-factor of the dark exciton in MoS₂ and the splitting between the dark and grey exciton. Overall, I feel this work lacks the novelty and importance that are required for the publication in Nature Communications. A similar method has been used to identify dark exciton in WSe₂ in 2017 (ref. 14), and the study on dark exciton has developed significantly since then, by coupling the dark exciton to a plasmonic substrate (ref.15), AFM tip. And various methods have been developed to directly imaging the dark exciton in WSe₂. Recently, the in-plane field method has also been used to identify the dark exciton in MoSe₂ (ref. 20). This work might be better suited for a specialized journal such as 2D materials, as the finding of the dark exciton in MoS₂, if confirmed, will be of interest to a specific group for the completion of the dark exciton study in TMDs. There are also a few issues that the authors need to address.

(1) The major one is the lack of TRPL data. Although the in-plane field dependence of the new peak is consistent with what has been reported for the dark exciton in WSe₂, the g factor is significantly smaller than that in WSe₂ and MoSe₂. As the author discussed, the lower dark exciton energy than the bright exciton contradicts the temperature dependence of the PL. Lifetime measurement of the dark exciton would help to confirm the dark exciton nature.

(2) The authors keep saying that the linewidth of the PL is too broad for extracting information directly. What is the linewidth for the dark exciton and bright exciton here? The authors reported PL linewidth as narrow as 2 meV in 2017 (ref. 28), and I would expect the dark exciton linewidth would be even smaller.

(3) For the MoTe₂ discussion, the authors attribute the absence of the dark exciton PL to the large energy difference between the dark and bright exciton, hence the large population difference. This is not entirely true, as the dark exciton typically has a much longer lifetime, which might still make the PL visible.

(4) There are a few typos. In page 4, it should be "As we cannot extract the energy of the 4 lines at weak magnetic field due to their linewidths". There is a missing "to". In page 6, it should be "CW" instead of "cw". There are a few missing commas and extra spaces. Overall, the writing of the paper could be improved. For example, in the method section, this sentence reads awkwardly: "High-quality MoS₂ and MoSe₂ MLs encapsulated in hexagonal boron nitride (hBN) and transferred onto an SiO₂ (80 nm)/Si substrate have been fabricated."

Reviewer #2 (Remarks to the Author):

In their manuscript the authors present the results of a magneto-optical study of MoS₂ and MoSe₂ monolayers focused on the elucidation of the exciton fine structure in these materials. The main observations are emerging emission signatures and characteristic splitting of the spectral features under strong in-plane and tilted magnetic fields. The data is further analyzed in the framework of an analytical model, describing the mixing of the exciton states in external magnetic fields accounting for the complete set of observations and allowing for the extraction of the relevant finestructure splitting parameters. The experimental data are very clear, the interpretations are strongly supported and the manuscript is overall very well presented. Moreover, the results of this study clarify a long-standing controversy of the exciton finestructure splitting in Mo-based materials and in MoS₂ in particular. They are an important addition to the initial reports on W-

based systems and thus allow, for the first time, for a complete experimental and theoretical picture of the exciton finestructure in van der Waals monolayer semiconductors. Due to the high relevance for a number of optical and transport processes, including direct consequences for spin-valley phenomena and emission efficiencies, the paper should find a broad audience in the community. I thus strongly recommend it for publication in Nature Communications journal. There are only a few points outlined in the following that the authors may consider to address.

- While the physics determining the complex mixing behavior of the exciton states in tilted B-fields are clear, the presentation still leaves room for some potential misinterpretation. As the authors state, the labeling in the key figure 3 illustrates dominant contributions of zero-field states to the 30T mixtures. The way it is currently phrased, however, combined with the solid lines that the eye then follows back to 0T in the figure, can easily imply that those are the states that the lines initially evolve from. Then one could, for example, think that the lower XB and XD should be labeled opposite (even if that is not the case as presented in SI7). The solution may be to include a few more details of what the solid lines mean directly into the manuscript and figure caption (+ potentially color the labels accordingly). May be there is even a convenient graphical way to indicate the mixing in the curves itself, or at least present the individual weights explicitly in the SI. Moreover, while the XB components in Table S1 and S2 are indeed dominant, the contributions XG and XD are rather similar. While mathematically correct, the small difference may not warrant the exclusive labeling, so that something like X_{G,D} and X_{D,G} may potentially better represent the actual weight distribution. Altogether, I admit to feel sufficiently confused about this part even writing the above, that hopefully further supports the point that the nice and interesting physics involved may find a slightly better representation for this subsection.

- From the paragraphs in the discussion I understand that the authors tried to detect the XG state in MoS₂ with high NA objectives. Would it be possible to suppress the low-NA emission from bright XB state for a higher chance to detect high NA luminescence from XG (may be that was tried as well though)? The reason is that, naively, one would expect that the observed splitting of 14 meV should be more than sufficient to ensure 99.99...% population of the dark state in MoS₂ at 5K (kBT ~ 0.5 meV), at least in thermal equilibrium. Then, as the authors discuss, would it mean that the oscillator strength of XG in MoS₂ is that much lower than in W-based materials (probably by orders of magnitude then, considering very strong signals for WSe₂, e.g.)?

- In the discussion/conclusion section I generally support the impact and implications of MoS₂ being a "dark" semiconductor with sizable bright-dark splitting in the overall context of quantum yield. However, it may be reasonable to soften at least a little bit the current phrasing considering the interpretation of temperature-dependent quantum yields or, in particular, the room temperature 100%-yield study. While there are clear implications for the bright state occupation, the interplay with non-radiative processes is likely to be one of the dominant factors for observed emission efficiency in many cases. It thus seems generally reasonable, e.g., indeed not to regard the decrease of PL in MoS₂ as clear distinction of a "bright"/"dark" material for which the present study offers the most convincing clarification. Also, if the material is indeed completely free from non-radiative traps (which is very unlikely in most cases of course), the redistribution of excitation in bright and dark states would only affect the total emission rate, but still keep the yield at 100%.

- Similar applies to the paragraph discussing spin-valley polarization in MoSe₂. I agree that the values are usually much lower in MoSe₂, but it also seems to very strongly depend on the excitation conditions. For resonant exciton, e.g., polarization close to 100% was in fact reported even for MoSe₂ (Tornatzki et al. PRL 121, 167401). The arguments of faster depolarization given by the authors are certainly still valid, yet I would only recommend to adjust the statement concerning weak polarization dependence of MoSe₂ to include "at non-/quasi-resonant conditions" and also cite the above paper for a more complete picture.

Reviewer #3 (Remarks to the Author):

The manuscript presents the magneto-photoluminescence measurement of the fine structures of exciton, including both spin-allowed bright exciton and spin-forbidden dark one, in MoS₂ and MoSe₂ monolayers with the application of very high magnetic field up to 30 Tesla that can be set to be in either the in-plane or tilted directions. Under the orientation-controllable high magnetic fields, rich fine structures of exciton measured optical spectra of exciton were revealed and, with the fitting of model simulation, the g-factors of bright and dark exciton states can be deduced. As a main finding stressed by the authors, the spin-forbidden dark exciton (SFDX) states of a MoS₂ monolayer was for the first time observed and shown to be below the bright one by $\Delta = 14\text{meV}$. The experimental studies are interesting and significant for better knowing the low-lying exciton fine structures of Mo-based TMD-MLs. However, I have the following comments and questions.

(1) To interpret the energy location of the SFDX, the authors did the analysis for fine structure splitting Δ , composed of the conduction band splitting Δ_{so} , binding energy difference between the SFDX and BX Δ_{bind} , and the e-h exchange interaction Δ_{exch} . With the estimated $\Delta_{so} \sim -3\text{meV}$ and $\Delta_{bind} \sim +8\text{meV}$, $\Delta_{exch} \sim +9\text{meV}$ is inferred. There exist however some uncertainties in the determination of the energy parameters. Beside DFT-model-dependent Δ_{so} , the Δ_{bind} should also somehow depend on the dielectric screening that could be modelled in different ways and depends on the chosen parameters. With the fixed Δ_{so} , the estimated value of Δ_{bind} directly determines the value of e-h exchange interaction energy. Can the author give more concrete details on the modelling of dielectric screening in the estimation of Δ_{bind} , and explain how is the modelling way suited for the real measured sample? With the estimated $\Delta_{bind} \sim +8\text{meV}$, the inferred Δ_{exch} is quite small as compared with the values reported by their previous paper, PRB 93, 121107. Can the authors justify the inferred e-h exchange interaction energy by comparing that with more other existing studies?

(2) Fig2b and Fig3c show the bright and dark exciton levels of MoSe₂-ML ordered in energy differently under the action of the in-plane and tilted magnetic fields, respectively. The level crossing around 5 T seen in Fig3c might lead to the reorder of the bright and dark exciton levels. Carefully examining the raw experimental data of Fig. 3a, more complex energy- and intensity-fluctuating features are observed over the wide range from 5 to about 15 T. For more understanding and interpretation of the rich spectral data of Fig.3a, can the model calculation based on Eqs S3-S5 be extended to simulate the spectral intensities of the exciton states, and provide more physical insight into experimental data?

(3) The data of MoS₂ under tilted field shown in Fig4 is quite ambiguous to identify the dark exciton signals. In particular, the grey exciton XG signal with surrounding spectral fluctuations in Fig4a is hardly recognized. The ambiguity in the identification of the exciton energy levels make the deduced g-factor from Fig4b questionable. Also, as one of the series of data of magneto-PL of Mo-based TMD-MLs, Fig4a should be presented in the same fashion as Figs 1a-3a so that reader can have the quantitative understand of the all cases on the equal footing. In the current format, the spectral data of other exciton states at low field are missing to see. If the signals of the dark states are intrinsically weak in the case of tilted field, the model simulation should be used to explain it.

(4) A schematic of experimental set-up and the complete PL spectra of the samples (even at $B=0$) over the wider spectral range showing the signals of all exciton complexes (e.g. trion, defect states, etc) could be added to the main article or the supplementary material to increase readers' recognition on the experiment and samples.

Overall, the manuscript presents a meaningful and interesting observation of the fine structures of exciton in Mo-based monolayer materials with the application of very high magnetic field. However, the attraction of the work mainly lies in the aspect of technology. Besides the advances in the employed magneto-PL technology, more physics behind the measured data should be explored

and interpreted with more insight to increase the value of the work. The manuscript could be considered to be published in this journal only if the physical or conceptual aspects of the work can be more enriched.

Reviewer #1 (Remarks to the Author):

“In this work, Robert et al. studied the PL from monolayer MoS₂ and MoSe₂ in the in-plane magnetic field or tilted magnetic field. They found an emerging peak in MoS₂ and MoSe₂ with the in-plane magnetic field, which they attribute to the dark exciton. Further application of the tilted magnetic field also enabled authors to extract the g-factor of the dark exciton in MoS₂ and the splitting between the dark and grey exciton. Overall, I feel this work lacks the novelty and importance that are required for the publication in Nature Communications. A similar method has been used to identify dark exciton in WSe₂ in 2017 (ref. 14), and the study on dark exciton has developed significantly since then, by coupling the dark exciton to a plasmonic substrate (ref.15), AFM tip. And various methods have been developed to directly imaging the dark exciton in WSe₂. Recently, the in-plane field method has also been used to identify the dark exciton in MoSe₂ (ref. 20). This work might be better suited for a specialized journal such as 2D materials, as the finding of the dark exciton in MoS₂, if confirmed, will be of interest to a specific group for the completion of the dark exciton study in TMDs.”

Answer:

We thank the reviewer for his/her time reading our manuscript. We perfectly agree that dark excitons in monolayers of WSe₂, WS₂ and MoSe₂ have been experimentally observed in the references that we mention in the paper. This was indeed not hidden in our first version of our manuscript as we cited all of them (to the best of our knowledge). But none of them unambiguously solved the problem for MoS₂ although this material is the most studied among the TMD family (for instance, ~1177 papers published in the past 12 months for MoS₂, compared to only 118 for WSe₂ or 94 for MoSe₂ – Source : Topics in WOS). We strongly believe that our work will be of importance for the broad audience of the TMD community and is not just a completion as each TMD material has very different properties (quantum yield, valley polarization ...). In MoS₂, even the sign of the bright-dark splitting was not known. Yet, as mentioned in the introduction of our manuscript the knowledge of the energy of dark excitons is of paramount importance for many reasons: it has a deep impact on the quantum yield as well as on spin relaxation mechanisms that govern the spin/valley properties in TMD. In particular, MoS₂ is the only material of the family exhibiting strong spin/valley polarization and intense photoluminescence at cryogenic temperatures (MoSe₂ polarization is very weak while the PL intensity of bright excitons in WSe₂ and WS₂ is less efficient at low temperature). Our work thus open the important questions of bright and dark exciton formation processes as well as relaxation.

“(1) The major one is the lack of TRPL data. Although the in-plane field dependence of the new peak is consistent with what has been reported for the dark exciton in WSe₂, the g factor is significantly smaller than that in WSe₂ and MoSe₂. As the author discussed, the lower dark exciton energy than the bright exciton contradicts the temperature dependence of the PL. Lifetime measurement of the dark exciton would help to confirm the dark exciton nature.”

Answer:

We agree with the reviewer that TRPL is generally a powerful tool to measure the lifetime of excitons as well as their relaxation processes. Unfortunately, performing TRPL at 30 Teslas is technically very challenging and to the best of our knowledge it has never been performed for a TMD monolayer. Indeed, both excitation and detection are fibered in our insert to avoid drift and vibration issues in the magnet. This is incompatible with the measurement of TRPL with good time resolution due to dispersion in fibers. We believe that the square dependence of the PL intensity presented in Fig. 1c as well as the measurement of the dark exciton g factor in tilted field already bring solid proofs of the dark exciton nature.

“(2) The authors keep saying that the linewidth of the PL is too broad for extracting information directly. What is the linewidth for the dark exciton and bright exciton here? The authors reported PL linewidth as narrow as 2 meV in 2017 (ref. 28), and I would expect the dark exciton linewidth would be even smaller.”

Answer:

We thank the reviewer for pointing out the linewidth issue. In our hBN/MoS₂/hBN samples a PL linewidth as narrow as 2 meV is indeed measured in a vibration free setup with a microscope objective with NA=0.82 (ref. 28) so that the size of the laser spot is diffraction limited (<1 μm). In the optical setup used in this work, the spot size is significantly larger (we estimate to be at best 2 μm), so that the linewidth is more sensitive to inhomogeneous broadening. For MoS₂, the linewidth of X_B is at best 7 meV while it is around 5 meV for X_D. The dark exciton linewidth is thus too broad to extract the small dark-grey splitting δ which is expected to be smaller than 1 meV. In MoSe₂ the homogeneity of the samples is better so that in pure Voigt geometry we obtain a linewidth of 1.4 meV for X_B and 2.4 meV for X_D/X_G. Again this linewidth is not sufficiently small to extract δ. Moreover, because X_D/X_G is at higher energy than X_B we can reasonably expect that the linewidth is not dominated by the radiative linewidth. In tilted field the excitation spot size is much bigger due to extended cross section of the laser spot with the tilted sample so that the linewidths are larger.

Changes in the text:

We have added in the text the linewidth of the transitions in both MoS₂ and MoSe₂ in the Voigt geometry measurements.

“(3) For the MoTe₂ discussion, the authors attribute the absence of the dark exciton PL to the large energy difference between the dark and bright exciton, hence the large population difference. This is not entirely true, as the dark exciton typically has a much longer lifetime, which might still make the PL visible.”

Answer:

We agree with the reviewer that dark excitons have a longer radiative lifetime than bright excitons. But if they lie at a higher energy than the bright excitons they are more likely to relax to bright states rather than radiatively recombining. Thus their lifetime is governed by the relaxation time rather than the radiative lifetime so that no PL is visible. This is the reason why we do not observe any PL from dark exciton in MoSe₂ at 0 T. The only way to observe PL from dark excitons when they lie at higher energy is to sufficiently mix them with bright states so that their radiative lifetime become comparable with the

relaxation time. This is only possible with high in-plane field and favored in MoSe₂ as the bright-dark splitting is small. As soon as the bright-dark splitting increases (as expected in MoTe₂) the mixing is much smaller. For instance in MoSe₂ a bright-dark splitting of 1.4 meV gives according to the model presented in the SI a brightening of 34% of the dark states at 30 T while it would be only 0.8% if the dark-bright splitting is 20 meV. This is why we propose that the dark exciton in MoTe₂ is at significantly higher energy than the bright state.

“(4) There are a few typos. In page 4, it should be “As we cannot extract the energy of the 4 lines at weak magnetic field due to their linewidths”. There is a missing “to”. In page 6, it should be “CW” instead of “cw”. There are a few missing commas and extra spaces. Overall, the writing of the paper could be improved. For example, in the method section, this sentence reads awkwardly: “High-quality MoS₂ and MoSe₂ MLs encapsulated in hexagonal boron nitride (hBN) and transferred onto an SiO₂ (80 nm)/Si substrate have been fabricated.””

Answer:

We thank the reviewer for pointing out these typos and we corrected them. We also modified the sentence in the methods section.

Reviewer #2 (Remarks to the Author):

“In their manuscript the authors present the results of a magneto-optical study of MoS₂ and MoSe₂ monolayers focused on the elucidation of the exciton fine structure in these materials. The main observations are emerging emission signatures and characteristic splitting of the spectral features under strong in-plane and tilted magnetic fields. The data is further analyzed in the framework of an analytical model, describing the mixing of the exciton states in external magnetic fields accounting for the complete set of observations and allowing for the extraction of the relevant fine structure splitting parameters. The experimental data are very clear, the interpretations are strongly supported and the manuscript is overall very well presented. Moreover, the results of this study clarify a long-standing controversy of the exciton fine structure splitting in Mo-based materials and in MoS₂ in particular. They are an important addition to the initial reports on W-based systems and thus allow, for the first time, for a complete experimental and theoretical picture of the exciton fine structure in van der Waals monolayer semiconductors. Due to the high relevance for a number of optical and transport processes, including direct consequences for spin-valley phenomena and emission efficiencies, the paper should find a broad audience in the community. I thus strongly recommend it for publication in Nature Communications journal. There are only a few points outlined in the following that the authors may consider to address.”

Answer:

We thank the reviewer for his/her very positive feedback.

“While the physics determining the complex mixing behavior of the exciton states in tilted B-fields are clear, the presentation still leaves room for some potential misinterpretation. As the authors state, the labeling in the key figure 3 illustrates dominant contributions of zero-field states to the 30T mixtures. The way it is currently phrased, however, combined with the solid lines that the eye then follows back to 0T in the figure, can easily imply that those are the states that the lines initially evolve from. Then one could, for example, think that the lower XB and XD should be labeled opposite (even if that is not the case as presented in SI7). The solution may be to include a few more details of what the solid lines

mean directly into the manuscript and figure caption (+ potentially color the labels accordingly). May be there is even a convenient graphical way to indicate the mixing in the curves itself, or at least present the individual weights explicitly in the SI.

Moreover, while the XB components in Table S1 and S2 are indeed dominant, the contributions XG and XD are rather similar. While mathematically correct, the small difference may not warrant the exclusive labeling, so that something like X_G,D and X_D,G may potentially better represent the actual weight distribution. Altogether, I admit to feel sufficiently confused about this part even writing the above, that hopefully further supports the point that the nice and interesting physics involved may find a slightly better representation for this subsection."

Answer:

We perfectly agree that the labelling of Figure 3 can lead to misunderstanding. We remove the labelling X_B , X_G and X_D at large field and replaced it with a labelling 1, 2, 3, 4 for the four mixed states. We keep the labelling X_B , X_G and X_D only at zero field in Fig.3c where it is correct. We also add a new figure (Fig.3d) to show the calculated weight of each component (X_B , X_G and X_D) in the four mixed states as a function of field. The size of the symbols is proportional to the weight of each component. This clearly show the strong mixing between bright and dark states especially for the two lowest energy states.

Changes in the text:

We changed the labelling in Figure 3 and 4 and added Fig.3d and Fig.4d.

"- From the paragraphs in the discussion I understand that the authors tried to detect the XG state in MoS2 with high NA objectives. Would it be possible to suppress the low-NA emission from bright XB state for a higher chance to detect high NA luminescence from XG (may be that was tried as well though)? The reason is that, naively, one would expect that the observed splitting of 14 meV should be more than sufficient to ensure 99.99...% population of the dark state in MoS2 at 5K ($k_B T \sim 0.5$ meV), at least in thermal equilibrium. Then, as the authors discuss, would it mean that the oscillator strength of XG in MoS2 is that much lower than in W-based materials (probably by orders of magnitude then, considering very strong signals for WSe2, e.g.)?"

We thank the reviewer for asking this very pertinent question. Indeed, we tried to detect the X_G state in MoS2 using high NA objective (NA=0.82) but we did not detect any features 14 meV below the bright transition. We even use the Fourier plan technique we employed in our previous work (supplement material of Phys. Rev. Lett. 119, 047401 (2017)) to enhance the visibility of z-polarized transitions. As shown in Fig. R1, we did not detect any grey exciton. As mentioned in the discussion section, this can be interpreted as a very weak oscillator strength for grey exciton in MoS2 or a very low population of this state at 5K. The issue of thermal equilibrium between bright and gray/dark state is questionable in MoS2 as the bright-dark splitting we measure (14 meV) is smaller than the Γ_5 optical phonon (36 meV) that permit to relax from bright to dark exciton (Reference 3 of the main text).

Changes in the text:

We added explicitly the value of the Γ_5 phonon in the text and added Figure R1 in the supplement material.

“- In the discussion/conclusion section I generally support the impact and implications of MoS₂ being a “dark” semiconductor with sizable bright-dark splitting in the overall context of quantum yield. However, it may be reasonable to soften at least a little bit the current phrasing considering the interpretation of temperature-dependent quantum yields or, in particular, the room temperature 100%-yield study. While there are clear implications for the bright state occupation, the interplay with non-radiative processes is likely to be one of the dominant factors for observed emission efficiency in many cases. It thus seems generally reasonable, e.g., indeed not to regard the decrease of PL in MoS₂ as clear distinction of a “bright”/“dark” material for which the present study offers the most convincing clarification. Also, if the material is indeed completely free from non-radiative traps (which is very unlikely in most cases of course), the redistribution of excitation in bright and dark states would only affect the total emission rate, but still keep the yield at 100%.”

Answer:

We perfectly agree with the reviewer and that was exactly the point we wanted to raise in the discussion section. The temperature dependence of PL intensity cannot be used to classify a TMD material as a “bright” or a “dark” material. Concerning our discussion on the 100% yield in super-acid treated MoS₂ we agree that if we consider a hypothetical material free from non-radiative traps and an infinite dark exciton lifetime the yield would still be 100%. We thus removed this small discussion.

Changes in the text:

We clarified the discussion part on the interpretation of temperature dependence measurements and remove the discussion on the super-acid treated samples.

“- Similar applies to the paragraph discussing spin-valley polarization in MoSe₂. I agree that the values are usually much lower in MoSe₂, but it also seems to very strongly depend on the excitation conditions. For resonant exciton, e.g., polarization close to 100% was in fact reported even for MoSe₂ (Tornatzki et al. PRL 121, 167401). The arguments of faster depolarization given by the authors are certainly still valid, yet I would only recommend to adjust the statement concerning weak polarization dependence of MoSe₂ to include “at non-/quasi-resonant conditions” and also cite the above paper for a more complete picture.”

Answer:

We agree that the problem of polarization measurement in MoSe₂ under quasi-resonant excitation is still under debate. In the reference cited by the referee (PRL 121, 167401), the polarization of MoSe₂ is measured on the trion and not the neutral exciton. Our discussion only relies on the fine structure of the neutral exciton. The fine structure of trions and its consequence on the spin/valley polarization definitely requires additional investigations. Also we want to point that the measurements performed by Tornatzky et al. are on non-encapsulated samples whereas the dark-bright splitting of 1.4 meV measured in our work are on hBN-encapsulated samples. As discussed, one ingredient of the bright-dark splitting is the exchange interaction which depends on the dielectric environment. We thus expect that in non-encapsulated samples (deposited on SiO₂), the exchange is stronger than in encapsulated samples and eventually turn the dark exciton below the bright exciton giving rise to stronger polarization. Of course, this is only speculative and measurement of dark-bright splitting as a function of dielectric environment will be required.

Changes in the text:

We moderate our statement concerning the measurements of polarization in MoSe₂ and added the reference proposed by the reviewer for completeness.

Reviewer #3 (Remarks to the Author):

“The manuscript presents the magneto-photoluminescence measurement of the fine structures of exciton, including both spin-allowed bright exciton and spin-forbidden dark one, in MoS₂ and MoSe₂ monolayers with the application of very high magnetic field up to 30 Tesla that can be set to be in either the in-plane or tilted directions. Under the orientation-controllable high magnetic fields, rich fine structures of exciton measured optical spectra of exciton were revealed and, with the fitting of model simulation, the g-factors of bright and dark exciton states can be deduced. As a main finding stressed by the authors, the spin-forbidden dark exciton (SFDX) states of a MoS₂ monolayer was for the first time observed and shown to be below the bright one by $\Delta = 14\text{meV}$. The experimental studies are

interesting and significant for better knowing the low-lying exciton fine structures of Mo-based TMD-MLs. However, I have the following comments and questions.”

Answer:

We thank the reviewer for his/her careful reading and his/her positive feedback.

“(1) To interpret the energy location of the SFDX, the authors did the analysis for fine structure splitting Δ_{so} , composed of the conduction band splitting Δ_{so} , binding energy difference between the SFDX and BX Δ_{bind} , and the e-h exchange interaction Δ_{exch} . With the estimated $\Delta_{so} \sim -3\text{meV}$ and $\Delta_{bind} \sim +8\text{meV}$, $\Delta_{exch} \sim +9\text{meV}$ is inferred. There exist however some uncertainties in the determination of the energy parameters. Beside DFT-model-dependent Δ_{so} , the Δ_{bind} should also somehow depend on the dielectric screening that could be modelled in different ways and depends on the chosen parameters. With the fixed Δ_{so} , the estimated value of Δ_{bind} directly determines the value of e-h exchange interaction energy. Can the author give more concrete details on the modelling of dielectric screening in the estimation of Δ_{bind} , and explain how is the modelling way suited for the real measured sample? With the estimated $\Delta_{bind} \sim +8\text{meV}$, the inferred Δ_{exch} is quite small as compared with the values reported by their previous paper, PRB 93, 121107. Can the authors justify the inferred e-h exchange interaction energy by comparing that with more other existing studies?”

Answer:

We thank the reviewer for this interesting question. Our estimation of $\Delta_{bind} = +8\text{ meV}$ does not rely on any complex modeling of the dielectric screening. We took the binding energy of the bright exciton measured in our previous work (Nat. Comm. 10, 4172 (2019)) for hBN/MoS2/hBN. $E_{binding_bright} = 221\text{ meV}$. Then we took the values of the calculated conduction band masses (0.46 and 0.43) and the valence band mass (0.6) from (2D Mater. 2 (2015) 022001). In a first approximation, the binding energy is proportional to the reduced exciton mass. So from a simple proportional law we obtain a binding energy for the dark exciton of $E_{binding_bright} = 229\text{ meV}$. Obviously, the main uncertainty in our estimation comes from the values of the effective masses taken from DFT calculations.

In our previous work (PRB 93, 121107), we compared the value of the spin-orbit splitting in the conduction band calculated with DFT+GW models with the bright dark splitting calculated by solving the Bethe Salpeter equation. In MoS2 we found a difference of around 15 meV between the two values. We want to point that this value corresponds to the sum of the exchange term Δ_{exch} and the difference of binding energies Δ_{bind} . The values estimated in this work $\Delta_{exch} = 9\text{ meV}$ and $\Delta_{bind} = 8\text{ meV}$ are thus consistent with our previous work. Finally, our estimation is also consistent with the value of $\Delta_{exch} = 6\text{ meV}$ calculated by Zhang and co-workers for WSe2 in the supplement material of Nature Nano 12, 883 (2017).

Changes in the text:

We clarified in the text how we estimate Δ_{bind} .

“(2) Fig2b and Fig3c show the bright and dark exciton levels of MoSe2-ML ordered in energy differently under the action of the in-plane and tilted magnetic fields, respectively. The level crossing around 5 T seen in Fig3c might lead to the reorder of the bright and dark exciton levels. Carefully examining the raw experimental data of Fig. 3a, more complex energy- and intensity-fluctuating features are observed

over the wide range from 5 to about 15 T. For more understanding and interpretation of the rich spectral data of Fig.3a, can the model calculation based on Eqs S3-S5 be extended to simulate the spectral intensities of the exciton states, and provide more physical insight into experimental data?"

Answer:

This question is related to the first question of Reviewer #2. In tilted field, the dark states and bright states of MoSe₂ undergo crossing and anti-crossing. To avoid misunderstanding, we removed our previous labelling X_B, X_G, X_D which is only strictly valid at zero field. As soon as magnetic field is applied, the four states are mixed. We add the Fig.3d to better show this mixing. Simulating the spectral intensities of the exciton states is indeed a good idea. Unfortunately, this requires to assume a thermal equilibrium between the four states which is highly questionable at low temperature. Secondly, we don't know the oscillator strength of the grey exciton. So we cannot calculate the contribution of light emitted with z-polarization.

"(3) The data of MoS₂ under tilted field shown in Fig4 is quite ambiguous to identify the dark exciton signals. In particular, the grey exciton XG signal with surrounding spectral fluctuations in Fig4a is hardly recognized. The ambiguity in the identification of the exciton energy levels make the deduced g-factor from Fig4b questionable. Also, as one of the series of data of magneto-PL of Mo-based TMD-MLs, Fig4a should be presented in the same fashion as Figs 1a-3a so that reader can have the quantitative understand of the all cases on the equal footing. In the current format, the spectral data of other exciton states at low field are missing to see. If the signals of the dark states are intrinsically weak in the case of tilted field, the model simulation should be used to explain it."

Answer:

We agree that the signal of dark states for MoS₂ in tilted field is weak but this is expected from the model. As shown in the Table S2, the contribution of bright components in the lowest energy states is only around 1% at 30 T. Moreover these weak transitions are blurred by the intense low energy tail of the brighter states. Nevertheless, we reproduced these features on other spots of the sample which give us confidence about the validity of the data. As suggested by the reviewer, we moved the color plot figure from the supplement material to the main text so that Fig. 3 for MoSe₂ and Fig.4 are now presented in the same way.

Changes in the text:

We modified the presentation of data in Fig. 4 in accordance with the one presented in Fig.3.

"(4) A schematic of experimental set-up and the complete PL spectra of the samples (even at B=0) over the wider spectral range showing the signals of all exciton complexes (e.g. trion, defect states, etc) could be added to the main article or the supplementary material to increase readers' recognition on the experiment and samples."

Answer:

Following the reviewer's suggestion we added a sketch of the experimental setup and PL spectra in a wider spectral range at 0 T in the supplemental materials

"Overall, the manuscript presents a meaningful and interesting observation of the fine structures of exciton in Mo-based monolayer materials with the application of very high magnetic field. However,

the attraction of the work mainly lies in the aspect of technology. Besides the advances in the employed magneto-PL technology, more physics behind the measured data should be explored and interpreted with more insight to increase the value of the work. The manuscript could be considered to be published in this journal only if the physical or conceptual aspects of the work can be more enriched.”

Answer:

We thank again the reviewer for his/her suggestions that helped us to improve the quality of our manuscript. We hope that the modifications we implemented now makes our manuscript suitable for publication in Nature Communications journal.

In conclusion, we thank the reviewers who helped us to enrich our manuscript. We believe that this new version now complies with the very high standards of Nature Communications.

Yours sincerely,

Dr. Cedric ROBERT (CNRS and University of Toulouse) on behalf of all co-authors,

REVIEWER COMMENTS

Reviewer #1 (Remarks to the Author)

I have thoroughly read the revised manuscript, SI, and authors' responses to all the reviewers. The authors have addressed some of my questions. However, after careful reading, I believe that the revised manuscript still does not meet the standard of Nature Communications for the three major reasons.

1. First, this work does not have enough novelty, and the result is not significant enough. The authors applied an in-plane magnetic field to brighten the dark exciton in MoS₂ and MoSe₂, similar to what has been done previously on WSe₂, WS₂, and especially MoSe₂. The work on MoSe₂ is not significantly different from previous work. The author showed that the dark exciton is the ground state for MoS₂. However, it is not clear why that would lead to new physics beyond what we have known for WSe₂, WS₂, and MoSe₂.
2. The experimental data quality shown in this work does not fully support the four-band theory presented in this work and should be significantly improved. Considering the significant improved quality of BN encapsulated TMD devices, including the authors' own efforts, the MoS₂ PL shown in this work is simply too broad to extract fine features needed for the authors' theory. The bright exciton PL does not even look like a single PL peak.
3. The results are inconsistent with what we know about MoS₂. For example, it is inconsistent with the temperature dependence of PL from MoS₂, including the authors' previous work (PHYSICAL REVIEW X 7, 021026 (2017)). The authors spend some efforts to reconcile the discrepancy. However, these explanations are based on speculation and hypothesis that are not supported by experiments or quantitative analysis. Some are even incorrect.

I think the manuscript is better suited for a specialized journal such as 2D Materials for PRB after the improvement. I list specific questions below.

1. In response to the authors' reply: "But none of them unambiguously solved the problem for MoS₂ although this material is the most studied among the TMD family (for instance, ~1177 papers published in the past 12 months for MoS₂, compared to only 118 for WSe₂ or 94 for MoSe₂ – Source : Topics inWOS)." The large number of papers on MoS₂ is mostly because the MoS₂ crystals are more abundant in nature and more accessible, which is also the reason why it was studied first among TMDs. This fact itself does not justify the novelty. The authors need to specify why the discovery of dark exciton in MoS₂ leads to new discovery compared with what has been done in WSe₂, WS₂, and MoSe₂.
2. Related to the authors' reply about the linewidth of the MoS₂, the PL spectra shown in Fig. 1 is quite broad and cannot even be described with a single Lorentzian peak at 0 T. There is at least one shoulder peak around ~1.94 eV (and even another one at ~ 1.95 eV). How do the authors determine the bright exciton peak position? Considering the

measured split is only on the order of 10 meV, accurate determination of the peak position and PL nature is critical.

3. Related to question (2), what is the doping of the MoS₂ studied and did the authors measure the PL spectra as a function of carrier density? This information will be important to identify different peaks. For example, dark trions would have the same dependence on the in-plane B field and will also have the same g-factor.
4. The authors need to show the line traces of the color plot in Fig. 2, especially for B field close to 10 T. It is not clear how the author could extract the dark exciton peak position at the field around 8 T. Fig.2a reproduced previous results of MoSe₂ (2D materials, 7 015017 2019), but the data is of lower quality. The dark exciton could not be seen until B field > 8 T (according to the authors), while it can be observed for the B field as low as 5 T in the previous work (2D materials, 7 015017 2019). The data extracted from Fig. 2a does not support the four-band theory developed by the authors. For example, the dark exciton behavior at the low B field transition to the grey exciton behavior at the high B field is a smooth change (Fig. 2b). At the intermediate B field, say 20 T, two PL peaks instead of one should be observed according to the four-band theory. However, this is not the case. Could the authors try to analyze the data from the previous work to see if that fits the authors' theory? The current data does not support this theory.
5. Also related to the fitting in Fig. 2b, if the fitting is not sensitive for $\delta < 1$ meV, why set it to be 0.6 meV instead of 0 meV? There is no guarantee that the value in WSe₂ (PHYSICAL REVIEW B **96**, 155423 (2017)) should be the same in MoSe₂, considering the sample difference and even the doping difference. Could the authors show the fitting result of 0 meV, is it better than the fitting with $\delta = 0.6$ meV? Did the author fit the data with the two-level model (just bright exciton and dark exciton) instead? The two-level model would also give four PL peaks for the tilted B field. Is the four-band fitting quantitatively better? The current PL data does not seem to support that the four-band theory instead of the two-level theory.
6. For Fig.3, the authors should also show the PL line trace for the low B field. Can the authors fit the data with just the Zeeman splitting of the dark and bright exciton? It is not clear that the current data is fitted better with the four-band theory.
7. What is the doping of the MoSe₂ studied here? Did the authors study the dependence on the carrier density, especially with an in-plane magnetic field, as shown in the previous work (2D materials, 7 015017 2019)?
8. Can the author show the PL line traces in Fig. 4 that shows peak 3 and 4? It is not clear how peak 4 can be extracted from the color plot. Again, can the authors fit the MoS₂ data with only the Zeeman splitting of the dark exciton and bright exciton, which will also give 4 peaks? Would the fitting be quantitatively better or worse than the fitting by the four-band theory?

9. The fact that the dark exciton is lower in energy than the bright exciton in MoS₂ is not consistent with the temperature dependence of the MoS₂, including the authors' previous work (PHYSICAL REVIEW X 7, 021026 (2017)). The author should perform temperature-dependent PL of MoS₂ in the presence of the in-plane B field. The PL intensity ratio between the dark exciton and bright exciton as a function of the temperature will tell the information about the energy difference between the dark exciton and bright exciton.
10. The authors claim that the reason they could observe dark exciton in MoS₂ with large NA object might be "We can speculate that either the oscillator strength of grey exciton is much smaller than in WX₂ and/or that their population remains weak despite lying at lower energy than the bright state." Do the authors have an estimation of the oscillator strength of the grey exciton compared with that of the WSe₂? Fine features of dark exciton such as dark trion and dark exciton phonon replica have been observed in WSe₂ (Nature Communications volume 11, Article number: 618 (2020); Nature Communications volume 10, Article number: 2469 (2019); Phys. Rev. Lett. 123, 027401, (2019); Phys. Rev. Research 1, 032007(R), (2019)). These fine features are supposed to have much smaller oscillator strength than the dark exciton of WSe₂ but still can be directly observed. Therefore, I am not convinced by the author's speculation.
11. The authors also claim, "For the population issue, we can notice that contrary to WX₂, the bright- dark exciton splitting in MoS₂ ML is smaller than the optical phonon energies which can lead to inefficient relaxation between bright and dark excitons". This speculation is incorrect. First of all, for non-resonant excitation, it is not correct to assume that the excited electron-hole pair to relax to the bright exciton state first. The exciton with excess energy can directly relax to the dark exciton state. Also, there is no guarantee the relaxation of the exciton to the dark exciton state is only via one particular optical phonon mode.
12. For the discussion of the g-factor of the dark exciton in MoS₂: "The measurements in the tilted field geometry presented below yield $gzD = -6.5$; this large value is in good agreement with the predicted one in a simple model²⁶ and the measured one in WSe₂ MLs^{21,27} (see table I)." The authors should explicitly include the theoretically predicted value here and compare it with the experimental data. Also, the authors should explain why the value of 6.5 for MoS₂ is a good agreement with the value of 9.4 (experimentally obtained from WSe₂).
13. Table I should include the previous results of MoSe₂ for completeness.
14. The authors claim that their results will be important for understanding the "trion fine structure." Could the authors elaborate on that, and why is it different from what we have known from WSe₂?
15. Did the authors use the four-band theory to fit the data in Fig. 1? How does that compare with the fitting with the two-level model?

16. How does the dark exciton intensity of MoSe2 scale with B field? Can the authors analyze data from Fig. 2 similarly as Fig. 1?
17. I would hesitate to call the MoS2 device in this work “high quality”, considering the broad PL width and the shape of the PL. The authors claim that the bright exciton peak width is 7 meV in the reply. Could the author show the fitting of the data shown in Fig 1? It is not clear how could the MoS2 bright exciton PL at 0 T (Fig. 1) be fitted with a single peak. The author attributed the broad PL to inhomogeneity broadening, but it is simply too large for a 2-micrometer spot.

Reviewer #2 (Remarks to the Author):

The authors provided a detailed, comprehensive response to the previously raised questions and included appropriate changes to the manuscript and supplementary. I thus further emphasize the importance of the reported findings for the field and strongly recommend publication of this work in Nature Communications journal.

(I apologize for not appending the following question/suggestion to the initial report. This should be thus considered as fully optional at this stage:

- Considering intensity dependence of the XG/XB ratio for MoSe₂ – is it correct that one would not expect the quadratic dependence with the magnetic field (compared to MoS₂) due to the competition between field-induced brightening and changes in the population ratio due to the energy splitting? Can the extracted intensity ratio be reasonably modeled using the two components or may one observe non-equilibrium scenario in the experiment?)

Reviewer #3 (Remarks to the Author):

With the appropriate responses to the reviewers' reports and substantial improvements in the figures, model simulation, and data interpretations, I agree with the publication of the revised manuscript in Nature Communications.

We appreciate the positive feedback of Reviewer #1 on our last version of our manuscript. Except from his/her concern about the 4-band theory, we understand that he/she now finds our replies satisfying. Concerning the comparison between the 2-band/4-band theory we have now added the results of both fitting in the supplement information as suggested by the reviewer.

We also addressed the minor points raised by the reviewer:

1. In reply to comments 12, the authors discussed the g-factor of MoS₂. The authors should include some of the discussion in the main text for the readers' sake.

Answer:

We have added the discussion on the small values of g factors in the supplement information (SI10).

2. In reply to comments 5, the authors presented an argument on why the δ in WSe₂ can be used in MoSe₂. I would not agree with the argument. Even based on the authors' own argument, δ should be a function of the binding energy. Without knowing the exact doping of the MoSe₂ sample (the authors showed some data with local doping), one cannot assume the same binding energy. But this is a minor issue, as the revision of the 4-band theory discussion as mentioned in the major concern should fix this.

Answer:

We agree that the binding energy of the exciton may change with doping. Nevertheless, in the samples studied in this work as well as in the WSe₂ sample of ref 22, we expect the doping to be small ($<10^{11}$ cm⁻²) as the signature of trion is not seen in reflectivity. Thus we don't expect a strong variation of the exciton binding energy as compared to the values of ref 29. This is why in the SI9, we present the results of the fit with three values of δ : 0 meV (2-band model), 0.6 meV (experimental value measured in WSe₂) and 2 meV to show that such a large value is inconsistent with our experimental results.

3. In the future, I strongly suggest the authors perform the temperature dependence experiment suggests in comment 9. I agree with the authors that PL is complicated, and the numbers might not quantitatively agree. However, we are talking about the trend of temperature dependence, and it is hard to imagine that the trend will be completely reversed. Citing the authors' comments, although the early work in ref. 7 shows a different number, but it is the temperature dependence in ref. 7 that reveals the existence of the dark exciton in WSe₂, which can explain the unexpected temperature dependence of PL.

Answer:

We thank the reviewer for this suggestion. Indeed, temperature dependence measurements may be useful to study the relative population of dark and bright excitons.

4. I would strongly suggest the authors perform doping control in the future.

Answer:

We thank the reviewer for this suggestion. We agree that doping control is mandatory to study dark trions.

List of changes:

All changes in the main text are in blue.

“Fig. SI 9a show the results of the fit with three values of δ (0, 0.6 and 2 meV). In Fig. 2b we present the results of the fit for $\delta=0.6$ meV which corresponds to the value experimentally measured in WSe₂. As δ is due to short range exchange interaction which scales with the exciton binding energy, we do not expect the values to be strongly different between TMD materials as the exciton binding energies are roughly the same.”

“Figure SI 9b show the results of the fit with three values of δ (0, 0.6 and 2 meV).”

REVIEWER COMMENTS

Reviewer #1 (Remarks to the Author):

I appreciate the authors' efforts in addressing my questions in the response letter. I found the replies to most of the questions satisfying. However, I still have a few concerns before I can recommend publishing this work on Nature Communications.

The major concern is the interpretation of the 4-band theory. Due to the limitation of the linewidth, the fine splitting was not resolved at low magnetic field. Therefore, there is no direct evidence confirming the 4-band theory. The fact that the fitting is not sensitive for $\delta < 1$ meV means that the fitting is similar to the 2-band theory, which is the case for $\delta = 0$.

I want to clarify that I am not against the 4-band theory. Instead, I found it intriguing. However, objectively, the data in this manuscript does not support that the 4-band theory is better than the 2-band theory. Rather, I found the fittings by the 2-band theory in the reply are similar to the fittings by the 4-band theory shown in the manuscript. I would strongly encourage authors to revise the manuscript to clarify that, in this work, the 2-band theory also fits well with current data. The authors should include the fitting results by the 2-band theory in the main text or SI, so that the readers could compare them with the fitting results by the 4-band theory.

There are also a few minor concerns:

1. In reply to comments 12, the authors discussed the g-factor of MoS₂. The authors should include some of the discussion in the main text for the readers' sake.
2. In reply to comments 5, the authors presented an argument on why the δ in WSe₂ can be used in MoSe₂. I would not agree with the argument. Even based on the authors' own argument, δ should be a function of the binding energy. Without knowing the exact doping of the MoSe₂ sample (the authors showed some data with local doping), one cannot assume the same binding energy. But this is a minor issue, as the revision of the 4-band theory discussion as mentioned in the major concern should fix this.
3. In the future, I strongly suggest the authors perform the temperature dependence experiment suggests in comment 9. I agree with the authors that PL is complicated, and the numbers might not quantitatively agree. However, we are talking about the trend of temperature dependence, and it is hard to imagine that the trend will be completely reversed. Citing the authors' comments, although the early work in ref. 7 shows a different number, but it is the temperature dependence in ref. 7 that reveals the existence of the dark exciton in WSe₂, which can explain the unexpected temperature dependence of PL.
4. I would strongly suggest the authors perform doping control in the future.

Below, we provide a more detailed, point by point response to the specific questions of the reviewers.

Reviewer #1:

1. In response to the authors' reply: "But none of them unambiguously solved the problem for MoS2 although this material is the most studied among the TMD family (for instance, ~1177 papers published in the past 12 months for MoS2, compared to only 118 for WSe2 or 94 for MoSe2 – Source : Topics inWOS)." The large number of papers on MoS2 is mostly because the MoS2 crystals are more abundant in nature and more accessible, which is also the reason why it was studied first among TMDs. This fact itself does not justify the novelty. The authors need to specify why the discovery of dark exciton in MoS2 leads to new discovery compared with what has been done in WSe2, WS2, and MoSe2.

Answer:

The comment of lack of novelty was already raised by the reviewer in his first report. In particular, it focuses exclusively on the fact that similar measurements have already been performed in other materials of the family to refute the interest to study MoS2. First, let us emphasize that the results on MoSe2 published in reference 30 were surprising for most of the community as we did not think that the dark exciton was lying so close to the bright exciton in MoSe2. We were thus happy to reproduce these results and share them with the community to confirm that the dark exciton effectively lies 1.5 meV above the bright one and confirm its dark nature in tilted field experiments. Secondly, although the TMD family is composed of five binary materials, they exhibit very different electronic and optical properties often related to the sign of conduction band spin-orbit splitting or bright-dark exciton splitting. In particular, as already explained in our previous response, MoS2 is the only material of the family exhibiting strong spin/valley polarization and intense photoluminescence at cryogenic temperatures (MoSe2 polarization is very weak while the PL intensity of bright excitons in WSe2 and WS2 is less efficient at low temperature). Finally, the results demonstrated in this work shows that MoS2 is the only material where the dark exciton lies below the bright one but still cannot be seen in a zero-field experiment. As explained in the discussion section of the paper this striking property raises two

important physical questions: the spin mixing between bands that could be reduced in MoS2 and explain a weaker oscillator strength than in the other materials (important to know for spin/valley physics) and the dark exciton formation mechanisms that may be different in MoS2 (important to fix limits to quantum yield or for future studies on condensation of dark exciton). Currently fingerprints of ferromagnetic spin ordering and also first order phase transitions are reported in optical measurements in monolayer MoS2 in gated devices (Phys. Rev. Lett. 124, 187602 (2020), Nat. Nano. 14, 432 (2019)). Knowing the exact order of spin-allowed versus spin forbidden transitions is thus crucial to further explore this many-body physics.

2. Related to the authors' reply about the linewidth of the MoS2, the PL spectra shown in Fig. 1 is quite broad and cannot even be described with a single Lorentzian peak at 0 T. There is at least one shoulder peak around ~1.94 eV (and even another one at ~ 1.95 eV). How do the authors determine the bright exciton peak position? Considering the measured split is only on the order of 10 meV, accurate determination of the peak position and PL nature is critical.

Answer:

We have already explained in our previous response that the linewidth in MoS2 measured in this work (7 meV) cannot be as narrow as in our previous work simply due to the size of the laser spot. In the setup of this work the spot diameter is at best 2 μ m while the best linewidth (around 2meV) have been reported with diffraction limited spot size (below 1 μ m). Figure R1 shows a PL spectrum taken with a diffraction limited spot size on the same sample as the one of Fig. 1 and Fig.4 of the main text. The measured linewidth is 2.6 meV which is perfectly consistent with the state of the art. Despite the problem of inhomogeneity of MoS2 that broadens the linewidths, the 14 meV splitting is resolved without ambiguity as seen in Fig.1b. The lineshape of the bright exciton has some minor shoulders on the high energy side but clearly does not change with magnetic field. The bright exciton peak position is determined at the main peak maximum.

Figure R1: PL spectrum of the same MoS2 sample than in the main text but taken with a diffraction limited laser spot and showing linewidth consistent with the state of the art.

3. Related to question (2), what is the doping of the MoS2 studied and did the authors measure the PL spectra as a function of carrier density? This information will be important to identify different peaks. For example, dark trions would have the same dependence on the in-plane B field and will also have the same g-factor.

Answer:

The doping of the MoS₂ monolayer is very small as we barely see the trion signature in PL (see Figure R1) and it is absent in reflectivity. The sample is not gated so we cannot tune the doping density. In some spot positions we did observe somewhat stronger (due to locally higher doping, i.e. charge puddles) signatures that can be interpreted as the PL of the trion. We even observed some change in the lineshape of the trion with a low energy shoulder appearing at large field that could be interpreted as the dark trion but we think that the data are not conclusive enough to be shown in this work (see Figure R2) and not the focal point of our discussion.

Figure R2: PL of trion in MoS₂ as a function of in-plane magnetic field that could be interpreted as the magnetic brightening of the dark trion.

4. The authors need to show the line traces of the color plot in Fig. 2, especially for B field close to 10 T. It is not clear how the author could extract the dark exciton peak position at the field around 8 T. Fig. 2a reproduced previous results of MoSe₂ (2D materials, 7 015017 2019), but the data is of lower quality. The dark exciton could not be seen until B field > 8 T (according to the authors), while it can be observed for the B field as low as 5 T in the previous work (2D materials, 7 015017 2019). The data extracted from Fig. 2a does not support the four-band theory developed by the authors. For example, the dark exciton behavior at the low B field transition to the grey exciton behavior at the high B field is a smooth change (Fig. 2b). At the intermediate B field, say 20 T, two PL peaks instead of one should be observed according to the four-band theory. However, this is not the case. Could the authors try to analyze the data from the previous work to see if that fits the authors' theory? The current data does not support this theory.

Answer:

The line traces of Fig. 2a are shown in Figure R3a. The bright and dark exciton peak positions are extracted by fitting spectra by two Voigt peaks. The spectrum measured at 30T is first fitted and the lower field spectra are fitted recursively by taking the results of the previous fit as the initial parameters. The critical field at which the dark exciton could be seen depends on the spot position. For instance, in the data shown in Figure SI6a, the dark exciton can be seen at 5T similarly as in the work presented in 2D materials, 7 015017 2019. The reviewer seems to not believe in the four-band theory. We strongly disagree on the fact that not experimentally distinguishing the four states proves that the four-band theory is wrong. We decided to develop the four-band theory because it is physically more accurate than the 2-band theory which is a simplification. In particular, the dark-grey splitting is often forgotten in the

literature while this is predicted by the group theory and has been experimentally measured in WSe2 (ref 21-27). We hope that it will stimulate new experiments to find a way to measure more precisely this splitting. We already explained in the previous response and in the main text that we can only give an upper bound to the dark-grey splitting due to the broadened linewidth. In Figure R3b, we copied Fig.2b from the main text and added the results of the 2-band theory in red. Of course, the data can be reasonably fitted by the 2-band theory as well but this simpler model remains a rougher approximation.

Figure R3: (a) Line traces corresponding to Fig.2a of the main text. (b) Same than Fig.2b with fit with the 2-band theory (red lines).

5. Also related to the fitting in Fig. 2b, if the fitting is not sensitive for $\delta < 1$ meV, why set it to be 0.6 meV instead of 0 meV? There is no guarantee that the value in WSe2 (PHYSICAL REVIEW B 96, 155423 (2017)) should be the same in MoSe2, considering the sample difference and even the doping difference. Could the authors show the fitting result of 0 meV, is it better than the fitting with $\delta = 0.6$ meV? Did the author fit the data with the two-level model (just bright exciton and dark exciton) instead? The two-level model would also give four PL peaks for the tilted B field. Is the four-band fitting quantitatively better? The current PL data does not seem to support that the four-band theory instead of the two-level theory.

Answer:

In the case of $\delta=0$ the 2-band theory is identical to the 4-band theory. We already explained in the main text that the fit is not sensitive to values of $\delta < 1$ meV and that we chose to present the results for $\delta=0.6$ meV as measured in WSe2. As δ is due to short range exchange interaction which scales with the exciton binding energy, we do not expect the values to be strongly different between TMD materials as the exciton binding energies are roughly the same (see Nat. Comm. 10 4172 (2019)).

0. For Fig.3, the authors should also show the PL line trace for the low B field. Can the authors fit the data with just the Zeeman splitting of the dark and bright exciton? It is not clear that the current data is fitted better with the four-band theory.

Answer:

Please see also answers above. Given the small value of δ , the fit is only sensitive at small B-field (see Figure R4(a) where the 4 lines cannot be distinguished (see the line traces in Figure R4(b)).

Figure R4: (a) Same than Fig.3c with fit with the $\delta=0$ (red lines). (b) Line traces at low field.

7. What is the doping of the MoSe₂ studied here? Did the authors study the dependence on the carrier density, especially with an in-plane magnetic field, as shown in the previous work (2D materials, 7 015017 2019)?

Answer:

The sample is not gated so we did not study the doping dependence. The residual doping is low as the PL of the trion is much smaller than the PL of the bright exciton and we did not see any signature in reflectivity. Nevertheless, as shown in Figure R5, some transitions appear with field around 1.612 eV that could be interpreted as the bright and dark trions but we think that the data are not conclusive enough to be shown in this work

Figure R5: Same than Fig2a on a broader energy scale. A peak that can be attributed to the bright trion (T) is visible around 1.612eV. At high field this peak is a doublet which could be interpreted as the signature of the magnetic brightening of dark trion.

8. Can the author show the PL line traces in Fig. 4 that shows peak 3 and 4? It is not clear how peak 4 can be extracted from the color plot. Again, can the authors fit the MoS₂ data with only the Zeeman splitting of the dark exciton and bright exciton, which will also give 4 peaks? Would the fitting be quantitatively better or worse than the fitting by the four-band theory?

Answer:

The peak 3 and 4 are indeed difficult to distinguish due to the small g factor of the bright exciton in MoS₂. The Zeeman splitting for a tilted field of 30T is only around 2meV; i.e. smaller than our linewidth. This is why we can only extract the values of peak 4 above 25T. The line traces are presented in Figure R6a. The result of the fit with $\delta=0$ is shown as red lines in Figure R6b. As stated previously, only the fit at low field is sensitive to the small values of δ .

Figure R6: (a) Line traces of Fig4a. Black dashed lines are peak positions of peak 3 and 4. (b) Same than Fig.4c with fit with the $\delta=0$ (red lines).

9. The fact that the dark exciton is lower in energy than the bright exciton in MoS₂ is not consistent with the temperature dependence of the MoS₂, including the authors' previous work (PHYSICAL REVIEW X 7, 021026 (2017)). The author should perform temperature-dependent PL of MoS₂ in the presence of the in-plane B field. The PL intensity ratio between the dark exciton and bright exciton as a function of the temperature will tell the information about the energy difference between the dark exciton and bright exciton.

Answer:

We already mentioned in the text that “we cannot use this simple argument to distinguish between a “bright” and a “dark” material as our results prove that the dependence of PL intensity with temperature in MoS₂ may be the result of a complex relaxation scheme between bright and dark states as well as its interplay with non-radiative channels.” In other words the populations of bright and dark excitons are not in a thermal equilibrium. Although the experiment suggested by the reviewer is certainly interesting, it cannot be used to measure the bright-dark splitting more directly. Here published work on WSe₂ proves this point : In references [12-15], it was precisely measured by several groups that the bright-dark splitting is 40 meV while in the early work of reference 7, this splitting was claimed to be 30 meV based on the PL intensity ratio.

10. The authors claim that the reason they could observe dark exciton in MoS2 with large NA object might be “We can speculate that either the oscillator strength of grey exciton is much smaller than in WX2 and/or that their population remains weak despite lying at lower energy than the bright state.” Do the authors have an estimation of the oscillator strength of the grey exciton compared with that of the WSe2? Fine features of dark exciton such as dark trion and dark exciton phonon replica have been observed in WSe2 (Nature Communications volume 11, Article number: 618 (2020); Nature Communications volume 10, Article number: 2469 (2019); Phys. Rev. Lett. 123, 027401, (2019); Phys. Rev. Research 1, 032007(R), (2019)). These fine features are supposed to have much smaller oscillator strength than the dark exciton of WSe2 but still can be directly observed. Therefore, I am not convinced by the author’s speculation.

Answer:

We would like to recall that our attempt to explain the weak intensity of PL of dark exciton is in a discussion section and that we hope it can stimulate future works to confirm or infirm our scenarios. It’s really difficult to have an estimation of the oscillator strength of the grey exciton. In reference 19, DFT+BSE calculations with their well-known uncertainty give an oscillator strength at least three orders of magnitude lower than the bright exciton in MoS2. We don’t have similar calculations for WSe2. Experimentally, the dark exciton lifetime in WSe2 is a few hundreds of ps as compared to the ps lifetime of bright exciton. The dark trion lifetime is also around 1ns, i.e. not significantly longer than the dark exciton lifetime. These lifetimes are not necessarily governed by the radiative lifetime but this suggest that the oscillator strength of the dark trion is not necessarily orders of magnitude smaller than the oscillator strength of the dark exciton.

11. The authors also claim, “For the population issue, we can notice that contrary to WX2, the bright-dark exciton splitting in MoS2 ML is smaller than the optical phonon energies which can lead to inefficient relaxation between bright and dark excitons”. This speculation is incorrect. First of all, for non-resonant excitation, it is not correct to assume that the excited electron-hole pair to relax to the bright exciton state first. The exciton with excess energy can directly relax to the dark exciton state. Also, there is no guarantee the relaxation of the exciton to the dark exciton state is only via one particular optical phonon mode.

Answer:

Non-resonant excitation generate a population of hot excitons that could relax either to the light cone of bright states or relax to the dark branch. But still, a phonon with the good symmetry needs to be involved to relax from the bright to the dark branch. This is the case of zone center Γ_5 optical phonon. Because its energy is larger (smaller) than the dark-bright splitting in MoS2 (WSe2), we expect this relaxation channel to be inefficient (efficient).

12. For the discussion of the g-factor of the dark exciton in MoS2: “The measurements in the tilted field geometry presented below yield -6.5 ; this large value is in good agreement with the predicted one in a simple model²⁶ and the measured one in WSe2 MLs^{21,27} (see table I).” The authors should explicitly include the theoretically predicted value here and compare it with the experimental data. Also, the authors should explain why the value of 6.5 for MoS2 is a good agreement with the value of 9.4 (experimentally obtained from WSe2).

Answer:

The simple model of reference 26 consists in separating the contributions of spin, valley and orbital in the value of the g factor. If we take into account only the spin and the orbital contributions we should get $g_{XB}=-4$ and $g_{XD}=-8$ which are roughly the values obtained for WSe2. In MoS2 the valley contribution seems to be more important. Taking the experimental value of $g_{XB}=-1.8$ we get a valley contribution of

2.2. If we assume that the valley contribution is similar for g_{XD} we expect that $g_{\text{XD}}=-5.8$. Of course taking the same valley contribution for both bright and dark excitons is a very rough assumption. In particular, as the conduction band masses involved in these two states is different, we could expect that the valley contribution for the dark exciton is smaller (larger conduction band mass). Nevertheless, we want to recall that this model is oversimplified and that theoretically predicting the g factor in semiconductors require much more sophisticated approach (see for instance arXiv:2002.11646v2 (2020)).

13. Table I should include the previous results of MoSe2 for completeness.

Answer:

We have added the results of reference 20 in the table.

14. The authors claim that their results will be important for understanding the “trion fine structure.” Could the authors elaborate on that, and why is it different from what we have known from WSe2?

Answer:

Our measurement of a dark-bright splitting of 14 meV in MoS2 implies that the conduction band spin orbit splitting (A^{SO}) is likely to be a few meV either positive or negative. We thus expect that the trion fine structure would be different whether A^{SO} is positive or negative. We believe that our measurement would be useful to interpret the recent observation of a negative trion doublet in reference 32.

15. Did the authors use the four-band theory to fit the data in Fig. 1? How does that compare with the fitting with the two-level model?

Answer:

We calculated the weight of bright components ($+1/6$ and $-1/6$) for the four states (the two brightened dark states and the two darkened bright states) as a function of in-plane field. In Figure R7, we plot the ratio of this weight between the two brightened dark states and the two darkened bright states. As expected, it scales quadratically with in-plane field because the splitting between bright and dark does not change significantly with B in opposite to MoSe2. So both models can be used to fit the data of Figure 1.

Figure R7: (Black solid curve) Weight of bright components ($+1/6$ and $-1/6$) in the two brightened dark states divided by the weight of bright components ($+1/6$ and $-1/6$) in the two darkened bright states. This calculation with the four-band Hamiltonian is very close to a simple quadratic law in a two level model (red curve).

16. How does the dark exciton intensity of MoSe₂ scale with B field? Can the authors analyze data from Fig. 2 similarly as Fig. 1?

Answer:

This question is related to the request of Reviewer #2 (see the answer below).

17. I would hesitate to call the MoS₂ device in this work “high quality”, considering the broad PL width and the shape of the PL. The authors claim that the bright exciton peak width is 7 meV in the reply. Could the author show the fitting of the data shown in Fig 1? It is not clear how could the MoS₂ bright exciton PL at 0 T (Fig. 1) be fitted with a single peak. The author attributed the broad PL to inhomogeneity broadening, but it is simply too large for a 2-micrometer spot.

Answer:

Please see more detailed reply to question 2. The sample quality at the state-of-the-art is confirmed in Fig.R1.

Reviewer #2:

The authors provided a detailed, comprehensive response to the previously raised questions and included appropriate changes to the manuscript and supplementary. I thus further emphasize the importance of the reported findings for the field and strongly recommend publication of this work in Nature Communications journal.

(I apologize for not appending the following question/suggestion to the initial report. This should be thus considered as fully optional at this stage:

- Considering intensity dependence of the XG/XB ratio for MoSe₂ – is it correct that one would not expect the quadratic dependence with the magnetic field (compared to MoS₂) due to the competition between field-induced brightening and changes in the population ratio due to the energy splitting? Can the extracted intensity ratio be reasonably modeled using the two components or may one observe non-equilibrium scenario in the experiment?)

Answer:

We thank the reviewer for this question. Indeed a thermal equilibrium between bright and dark states is questionable. As reported in ref.20, the ratio between dark and bright exciton intensities in MoSe₂ does not follow a simple quadratic law as for MoS₂. The plot corresponding to our data of Fig.SI6a where the dark exciton can be observed at the lowest field is shown in Figure R8a. It is very similar to the results of Ref.20. where the saturation of this ratio and its decrease at high field has been interpreted as a change of the thermal population of the dark state with field as the splitting between bright and dark states increases with field. Using our model, we calculated the weight of bright components (+1

6 and -1 6) for the four states (the two brightened dark states and the two darkened bright states) as a function of in-plane field and plot the ratio between the two brightened dark states and the two darkened bright states as in Figure R7. The result is shown in Figure R8b (red curve). Interestingly, for MoSe₂, it differs from a simple B² law (black curve) because the splitting increases with B field. In Figure R8b, we also show this ratio after multiplication by a factor $\exp(-\Delta_{\text{bright-dark}}/kT)$ assuming a thermal equilibrium. A temperature of 16K would be required to match with our data.

(a)

(b)

Figure R8: (a) Ratio of PL intensities of dark and bright excitons extracted from Fig.S16. Calculation of the dark/bright intensities (black) simple B^2 law, (red) same than in Figure R7, (blue, pink green, navy blue) same than red but multiplies by $\exp(-\Delta_{\text{bright-dark}}/kT)$.

Reviewer #3:

With the appropriate responses to the reviewers' reports and substantial improvements in the figures, model simulation, and data interpretations, I agree with the publication of the revised manuscript in Nature Communications.

Answer:

We thank again the reviewer for his/her help improving our manuscript and his/her positive feedback.